# Identification of Crop Type in Crowdsourced Road View Photos with Deep Convolutional Neural Network

**DOI:** 10.3390/s21041165

**Published:** 2021-02-07

**Authors:** Fangming Wu, Bingfang Wu, Miao Zhang, Hongwei Zeng, Fuyou Tian

**Affiliations:** 1State Key Laboratory of Remote Sensing Science, Aerospace Information Research Institute, Chinese Academy of Sciences, Beijing 100101, China; wufm@radi.ac.cn (F.W.); zhangmiao@radi.ac.cn (M.Z.); zenghw@radi.ac.cn (H.Z.); tianfy@radi.ac.cn (F.T.); 2College of Resources and Environment, University of Chinese Academy of Sciences, Beijing 100049, China

**Keywords:** crop type, crowdsourced road view photo, deep convolutional neural network, automatic photo identification, ensemble classification

## Abstract

In situ ground truth data are an important requirement for producing accurate cropland type map, and this is precisely what is lacking at vast scales. Although volunteered geographic information (VGI) has been proven as a possible solution for in situ data acquisition, processing and extracting valuable information from millions of pictures remains challenging. This paper targets the detection of specific crop types from crowdsourced road view photos. A first large, public, multiclass road view crop photo dataset named iCrop was established for the development of crop type detection with deep learning. Five state-of-the-art deep convolutional neural networks including InceptionV4, DenseNet121, ResNet50, MobileNetV2, and ShuffleNetV2 were employed to compare the baseline performance. ResNet50 outperformed the others according to the overall accuracy (87.9%), and ShuffleNetV2 outperformed the others according to the efficiency (13 FPS). The decision fusion schemes major voting was used to further improve crop identification accuracy. The results clearly demonstrate the superior accuracy of the proposed decision fusion over the other non-fusion-based methods in crop type detection of imbalanced road view photos dataset. The voting method achieved higher mean accuracy (90.6–91.1%) and can be leveraged to classify crop type in crowdsourced road view photos.

## 1. Introduction

Zero Hunger has been recognized as one of the core sustainable development goals [1,2]. Although global food production is increasing, some countries have still been short of food in recent years [3]. Against the background of global climate change, the frequency of extreme weather further increases the uncertainty of food production. Timely, transparent, and accurate information on global agricultural monitoring is essential for ensuring the proper functioning of food commodity markets and limiting extreme food price volatility [4]. Accurate and reliable crop type information is vital for many applications such as crop area statistics, yield estimation, land use planning, and food security research.

Remote sensing techniques have been proven to be an efficient, objective, and cost-effective method of agricultural monitoring at global, national, and sub-national scales. With more remote sensing data being made public and the development of cloud computing, it is possible to use these data for the large-scale classification of farmland types [5,6,7,8]. However, influenced by environmental factors such as elevation distribution, farmland area, land cover richness and cloud cover frequency, the overall accuracy of the four farmland products was below 65% and the standard deviations among all four cropland datasets varied from 0 to 50% [9]. In order to improve the accuracy of future cropland products, cropland classification methods require more and richer training or verification data collected from ground surveys to build a robust model for crop identification [10].

The crowdsourcing method is available for collecting field data and managing public access (such as Geo-WiKi.org and iNaturalist.org). Crowdsourcing geo-tagged images from Flickr and Geograph were used to create a binary land cover classification (developed/undeveloped) for an area of 100 × 100 km^2^ in Great Britain, and the accuracy achieved was around 75% [11]. Roadside sampling strategies for cropland data collection that enable the sampling of large areas at a relatively low cost have been suggested [12,13] and integrated with remote sensing data to provide crop acreage estimations [14]. Quality control of crowdsourcing geo-tagged data is very important; otherwise, the user will be unable to assess the quality of the data or use it with confidence [15]. For crowdsourcing photo classification, visual interpretations by volunteers have been used in previous studies [16]. On the other hand, they also discussed the long time required for visual interpretation of many photos, and automatic approaches should be proposed instead of manual classification.

Deep learning is a recent, modern technique for photo processing and data analysis that has resulted from the continued development of computer hardware and the appearance of large-scale datasets [17]. Convolutional neural networks (CNNs), one of the most successful network architectures in deep learning methods, have been developed for photo recognition and applied to complex visual photo processing. Deep learning CNNs have entered the domain of agriculture, with applications such as plant disease and pest recognition [18], picking and harvesting automatic robots [19], weed-crop classification [20], and monitoring of crop growth [21]. Photo datasets are the most commonly used basic data in the field of deep learning. Some research data come directly from crowdsourced datasets, such as ImageNet [22], iNaturalist [23] and PlantVillage [24]. Most applied agriculture research collects sets of real photos based on the research needs of fine-grained photo categorization, such as DeepWeeds [25], CropDeep [26] and iCassava 2019 [27]. The DeepWeeds dataset consists of 17,509 labelled images of eight nationally significant weed species native to eight locations across northern Australia. The CropDeep species classification and detection dataset, consisting of 31,147 images with over 49,000 annotated instances from 31 different classes of vegetables and fruits grown in greenhouses. iCassava 2019 is a dataset consisting of 9436 labeled images covering healthy cassava leaves as well as 4 common diseases. These datasets do not focus on crop types in the field and all photos are close-ups of the identified objects.

However, if there are no public benchmark datasets specifically designed for crop type classification, this limits the further application of deep learning technology and the development of intelligence in acquiring accurate crop types, distributions, and proportions. A study in Thailand explored the potential of using deep learning to classify photos from Google Street View (GSV) for the identification of seven regionally common cultivated plant species along roads, and the overall accuracy of the multiclass classifier was 83.3% [28]. A total of 8814 GSV images with 7 classes of crop for the Central Valley and 1 class representing “other” were prepared for training the CNN model in the USA, and the overall classification accuracy was 92% [29]. Obviously, the timing of GSV photo capture may not be during the growing seasons, and the revisit frequency in most rural areas is low to nonexistent. It has been demonstrated that GSV survey detected fewer plants than car surveys in Portugal countrywide [30]. The dataset of the first study is not public, and, while the dataset of the second can be download freely, the size of the dataset is relatively small. In addition, more state-of-the-art classification networks and model ensemble could be compared and selected to improve performance.

The objectives of this paper are (1) to build a large road view crop photo dataset to support automatic fine-grained classification with deep learning, and (2) to identify and fuse the optimal deep learning architectures for road view photo classification.

The remainder of this paper is organized as follows: Section 2 presents a specific description of the iCrop datasets and introduces the deep learning classification network and the selected data augmentation process. Section 3 presents the experimental performance and results. Section 4 discusses which model is best for the different applications. Finally, we summarize the conclusions and propose our further research aims in Section 5.

## 2. Materials and Methods

### 2.1. Dataset

To build a road view crop photo dataset to support automatic fine-grained classification with deep learning, the crowdsourced photos were collected, cleaned, labeled and divided as shown in Figure 1.

#### 2.1.1. Data Collection

The data were collected by a smartphone app named GVG as part of a crowdsourcing project initiated by the CropWatch team since 2015 [31]. GVG is mobile phone software that can record the photos, location and time of crops at the same time, and users can mark the types of crops in the photos. It is freely downloaded from application marketplaces such as Google Play, the Apple App Store, the Huawei App Gallery and other app marketplaces. The GVG application is easy to use for non-professionals and reduces the amount of ground observation work. A tutorial of field data collection with GVG can be download from http://www.nwatch.top:8085/icrop/docs/gvg.pdf (accessed on 28 October 2020). As shown in Figure 2, the phone was mounted on the window for fast roadside sampling along the road with the help of vehicles. Hundreds of thousands of roadside view photos were collected automatically from the main grain-producing areas of China, including Liaoning, Hebei, Shandong, Jilin, Inner Mongolia, Jiangxi, Hunan, Sichuan, Henan, Hubei, Jiangsu, Anhui, and Heilongjiang. The sampling time was based on the crop phenology calendar. These data have been used to support the paddy field/dry land identification and other land cover mapping [6,32].

#### 2.1.2. Dataset Cleaning and Labeling

First, the photos without cropland or severely blurred field photos were deleted; then, all valid photos were annotated and submitted by the observers, consisting of photos, classes, locations and observation time. A web photo management system was built to easily view and manage the photos based on the Piwigo photo management tool. When a user logs in, they can check whether the photo classification is correct and mark the incorrectly classified photo with the correct classification. All users can rate the trustworthiness of the photo tags on a five-point scale. Photos with an average score less than 3 will be removed or re-tagged. With this method, 34,117 correct photos were divided into twelve types, representing the dominate crop types or farmland without crop, including cotton, maize, peanut, rape, rice, sorghum, soybean, sunflower, tobacco, vegetable, and wheat.

#### 2.1.3. Training and Validation Subsets

Based on fine annotation, the photos were randomly divided into a training set and a test set with approximately 80% of the photos included in the training set. The 80/20 split rate of the training/test dataset is the most common in deep learning applications, and other similar split rates (e.g., 70/30) should not have a significant impact in the performance of the developed model [33]. These empirical proportions make up for the imbalance problem in the dataset. Even if there are some classes with a large number of training samples, their corresponding test sets also contain more samples; thus, these samples undergo more rigorous assessment. At this point, we have the final photo splits, with a total of 27,401 training photos and 6716 test photos. In Table 1, we list the numbers of training and test photos in each category in the iCrop dataset. Randomly selected sample photos from each category of the dataset can be viewed in Figure A1 in Appendix A. It can be seen from the picture that the weather, color, and angle of each photo are different, and some crops are partially obscured by trees and buildings along the road.

The largest crop class was “rice”, with 5850 photos, and the smallest was “sorghum”, with 149 photos. We can see that the photos of each class are imbalanced, which represents a long-tailed real-world challenge in classification problems.

### 2.2. Method

The general structure of the presented method identifying and fusing optimal deep learning architectures for road view photo classification is shown in Figure 3.

#### 2.2.1. Data Augmentation

The data scale has a great influence on the accuracy of a deep learning network model. A small amount of data will lead to the overfitting of the model, making the training error very small and the testing error extremely large. To avoid overfitting of the network, some augmented preprocessing was applied to enhance a large number of photos in the dataset before training [34]. The augmentation process has been reported to improve classification accuracy in many studies [35,36,37,38,39,40]. We used two primary ways to generate new photos from raw photo data with very little computation before training, and the transformed photos only need to be stored in memory. The first method is geometrical transformations consisting of resizing, random cropping, rotation and horizontal flipping [41]. The second method is intensity transformations consisting of contrast, saturation, brightness, and color enhancement [42].

#### 2.2.2. Convolutional Neural Networks

Deep learning allows computing models from multiple processing layers to learn data that have multiple abstract levels. Although a series of CNN models have shown outstanding performance on plant disease detection and diagnosis, the challenges related to addressing other agricultural tasks online or offline are still difficult to overcome [43]. Therefore, to characterize the classification difficulty of iCrop, we ran experiments with five state-of-the-art CNN models, including Resnet50, InceptionV4, Densenet121, MobileNetV2 and ShuffleNetV2.

ResNet50, which was the champion of the ImageNet Large Scale Visual Recognition Challenge (ILSVCR) 2015, introduces a new residual structure and solves the problem that the accuracy rate decreases as the network deepens [44]. Once it was established, InceptionV4 was improved from InceptionV3 (the winner of ILSVRC2014) with resumed connectivity, greatly accelerating training and improving performance through ResNet’s structure [45]. The best paper of the IEEE Conference on Computer Vision and Pattern Recognition (CVPR) 2017 proposed a densely connected network structure named DenseNet121, which is more conducive to the transmission of information flow [46]. At present, most deep learning networks run on computers with strong floating-point computing power. However, MobileNetV2 and ShuffleNetV2 are designed for mobile and embedded visual applications. The finer tuning of MobileNetV2 based on the MobileNet structure, skipped linking directly on a thinner bottleneck layer, and no ReLu nonlinear processing on the bottleneck layer can achieve better results [47]. ShuffleNet-V2 is a lightweight CNN network that balances speed and accuracy [48]. At the same complexity, it is more accurate than ShuffleNet and more suitable for mobile and unmanned vehicles.

These networks were implemented in PaddlePaddle deep learning frameworks, which is an open-source platform with advanced technologies and rich features [49]. An excellent RMSprop optimizer proposed by Geoff Hinton was used in training the adaptive learning rate [50]. Training batches of size 30 were created by uniformly sampling from all available training photos as opposed to sampling uniformly from the classes.

#### 2.2.3. Ensemble Classification

Ensemble is the process of fusing information from several sources after the data have undergone preliminary classification to improve the final decision [51]. To improve crop identification accuracy, deep learning networks with good accuracy and fast speed will be selected to decision fuse. These models can be seen as different experts focusing on different point of views, whose decisions are complementary and could be fused as a more accurate and stable one. Ensemble classification based on majority voting is proposed in this paper. Majority Voting is one of the most popular, fundamental and straightforward combiners for the predictions from multiple deep learning algorithms [52]. Every individual classifier vote for one class label. The class label that most frequently appears in the output of individual classifiers is the final output. Majority voting was applied on the individual classification results of all classifiers without a reliability check.

## 3. Results

Experiments with deep learning classification architectures were carried out. All models were trained and tested on an Intel(R) Xeon(R) Gold 6148 CPU @ 2.40 GHz with NVIDIA Tesla V100-SXM2 GPU and 16G RAM. The training proceeded on the training set, after which the evaluation was performed on the validation set to minimize overfitting. When the training process and parameter selection were achieved, the final evaluation was performed on the unknown testing set to evaluate the performance. During training and testing, the photo was adjusted to 224 px as the input of the network.

### 3.1. Accuracy of Single CNN

The classification test accuracy across all species of each model is shown in Table 2. Classification accuracy is mentioned as the Rank-1 identification rate per class [53]. The percentage of correct predictions where the top class (the one with the highest probability), as indicated by the deep learning model, is the same as the previously annotated target label. For multiclass classification problems, “Average accuracy” indicates the total number of correct prediction samples divided by the total number of testing photos. We observe a larger difference in accuracy and small difference in a average accuracy across the different crops.

As seen in Table 2, for rapeseed, the DenseNet121 model outperformed the other models with an accuracy of 96.7%. However, for wheat, the MobileNetV2 model outperformed the other models with an accuracy of 96.0%. For average accuracy, the ResNet50 model outperformed the other models with an average accuracy of 87.9%. MobileNetV2 and ShuffleNetV2 have similar average accuracies of 87.5%. However, the worst model was DenseNet121, which obtained an accuracy of only 86%.

### 3.2. Identification Efficiency of Single CNN

The original intention of the collected database was to construct an intelligent platform that could be operated online or offline on various mobile phones and other equipment; this database required not only accuracy but also real-time performance to further improve the overall timeliness and efficiency of precision crop structures. Thus, the frames per second (FPS) were selected as the evaluation indicator to evaluate the speed performance of each classification model on the same machine. Please note that the time taken to perform the required preprocessing steps was also measured. These steps include loading a photo and resizing it for input to the network. The evaluation results of the detection time are shown in Figure 4. The result shows that ShuffleNetV2 with approximately 13 FPS is fastest to meet the needs for real-time cropland classification.

### 3.3. Fusion Accuracy

To avoid the draw problem, the number of classifiers performed for voting is usually odd. We had two voting schemes, one named voting-5 which contained five CNN classifiers and another named voting-3 which contained three CNN classifiers. As shown in Figure 5, the ResNet50 model outperformed the other models on average classification accuracy, and the ShuffleNetV2 model outperformed in classification speed. However, the differences among ResNet50, MobileNetV2 and ShuffleNetV2 on average accuracy and speed were very small. Therefore, the classification results of ResNet50, MobileNetV2 and ShuffleNetV2 were selected in the voting-3 scheme. According to the comparisons presented in Table 3 and Figure 5, small differences in accuracy and average accuracy can be observed between voting-5 and voting-3 schemes.

## 4. Discussion

We present a road view crop type dataset named iCrop for the development of crop type classification systems to support remote sensing crop distribution mapping as well as crop area estimation. To the best of our knowledge, no comparable, publicly available dataset exists that is the basis for deep learning research, and the datasets that are currently available (the most influential is ImageNet) do not have many photographs of arable crop, and the angles and distances of the shots are different. Therefore, the photos taken by GVG were sorted, classified and corrected, and the training set and test set were divided.

Unlike GSV, our photos were collected during the crop growing season, capturing the differences in the field as the places and mobile phones change. The angles, heights and directions of the photos are different for each person, and photos also vary in resolution, color, contrast, and clarity. The photos in the datasets contain similar characteristics and imbalance among each class, which reflects the long-tailed real-world challenges in classification problems. Therefore, our photos are more challenging to classify than GSV photos.

The baseline classification results were determined from our experiments. We can see that state-of-the-art CNN models have room to improve when applied to imbalanced roadside view crop datasets. None of the CNN models have the best recognition accuracy for all kinds of crops. The test data, training environment, iteration times and other conditions are the same, but the complexity of the structure for each model is different. The accuracy and complexity of the model are not necessarily related. For wheat, rice, tobacco, and sunflower lightweight models such as MobileNetV2 and ShuffleNetV2, the crop classification accuracy is high. There are different feature fusion methods between DenseNet121 and ResNet50, and the accuracies of classification for different kinds of crops are also similar, but in general, ResNet50 exhibits slightly higher accuracy. Different CNNs are “accurate” in certain aspects, so model fusion could improve the final prediction ability to a greater or lesser degree. According to the comparisons in Table 1 and Table 3, the ensemble classification accuracy is higher than individual models for most species, and the average accuracy is also higher than that of each model; in particular, voting with five classifiers increased the overall accuracies by up to 3.5%.

Figure 6 shows the normalized confusion matrix resulting from combining the voting-3 and voting-5 models’ performances across the 12 cross-validated test subsets. The model confuses 8% of bare land images with vegetable, and 3% vice versa. Reviewing these particular samples shows that bare land with weeds looks strikingly similar to cropland with small leafed vegetables, while some vegetable gardens contain small patches of bare land. This is illustrated in the sample misclassification of bare land in Figure 7a,b. This likeness is the reason for these false positives in our model. Figure 7c,d also show that crops badly shaded by trees or grasses in photo also affect the classification accuracy. It is necessary to add segmentation information to photos for deep learning to extract farmland types more accurately. Another way to protect crops from being obscured by other objects is to use drones to take pictures over farmland. Figure 6 and Table 1 illustrate that the accuracy rates are higher for rapeseed, which did not have the largest training sample size. This relatively high performance on distinct crop color features is likely because CNN exhibits better performance for capturing the color characteristics of these crop types.

The top test accuracies against the number of training photos for each class from the five classification models and ensemble models are plotted in Figure 8. It is shown that there is a positive correlation between the number of training images and the test accuracy. The consensus of most current studies is that for deep learning, the performances will increase with growing data size [54,55,56,57]. However, we still observe a variance in the accuracy for classes with a similar amount of training data, revealing opportunities for algorithmic and dataset improvements in both the low data and high data regimes. The ensemble classification accuracy is significantly higher than individual model for low data species. The training images of peanut, sunflower and sorghum are all lower than those of other crops, and the performances for these crops are even low after fusion. it is caused by dataset imbalance. No imbalance-correcting technique can match adding more training data when it comes to measuring precision and recall. We suggest corresponding data collection efforts for classes with few photo samples should be also underway use a hybrid method that fuses GSV images and crowdsourcing data.

## 5. Conclusions

In this paper, a first large, public, multiclass road view crop photo dataset named iCrop is established for the development of crop type detection with deep learning. The iCrop dataset contains 12 types, representing the most popular crop types or farmland without crops, using 34,117 photos, and outlines the baseline performance for state-of-the-art deep convolutional neural networks. The results show that DCNN has good potential application in crop type detection from road view photos, and these computer vision models have room to improve when applied to imbalanced crop datasets. Small efficient ShuffleNetV2 models designed for mobile applications and embedded devices have better real-time performance (13FPS) and average accuracy (87.5%).

The deep learning network models with good accuracy and fast speed were selected, and the major voting decision fusion method was used to improve crop identification accuracy. The results clearly demonstrate the superior accuracy of the proposed decision fusion over the other non-fusion-based methods in crop type detection of imbalanced road view photos dataset. The voting method achieved a mean accuracy of 91.1%, which can be leveraged to classify crop type in crowdsourcing road view photos online. The proposed fusion strategy increased overall accuracies by up to 3.5% compared to the best single CNN model.

We anticipate that our proposed method will save researchers valuable time that they would otherwise spend on the visual interpretation of larger number of photos. In the future, we plan to update the dataset and include more diverse crop types from worldwide areas to expand the scope of the iCrop. With the increasing use of drones in agriculture, drones can also be used as a tool to collect photos of crops to solve the problem of crops being obscured. Meanwhile, mobile technology has developed at an astonishing rate in the past few years and will continue to do so. With ever-improving computing performance and storage capacity on mobile devices, we consider it likely that highly accurate real-time crop classification via smartphones is just around the corner.

## Figures and Tables

**Figure 1 sensors-21-01165-f001:**
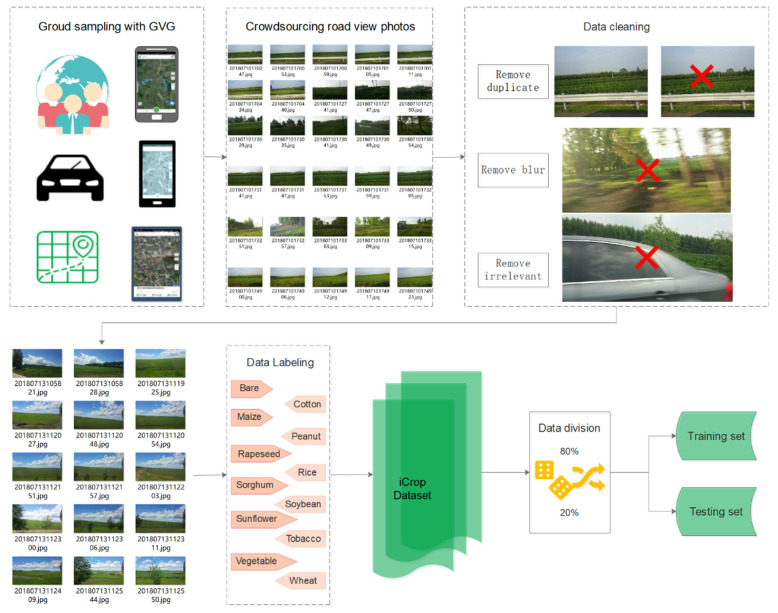
A workflow diagram of crowdsourced crop photos collection, cleaning, labeling and division.

**Figure 2 sensors-21-01165-f002:**
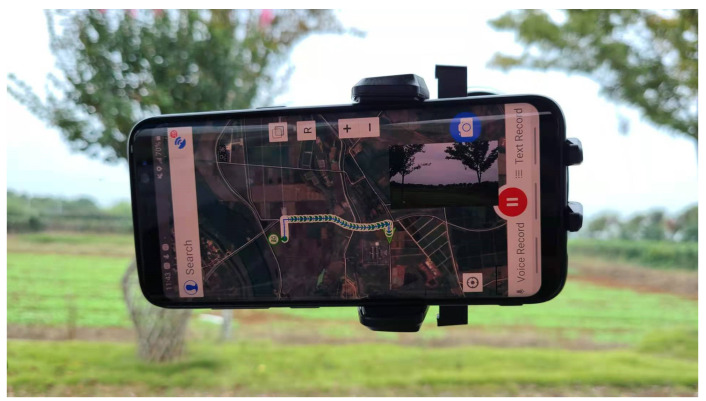
GVG was used for data collection along the road base from a vehicle.

**Figure 3 sensors-21-01165-f003:**
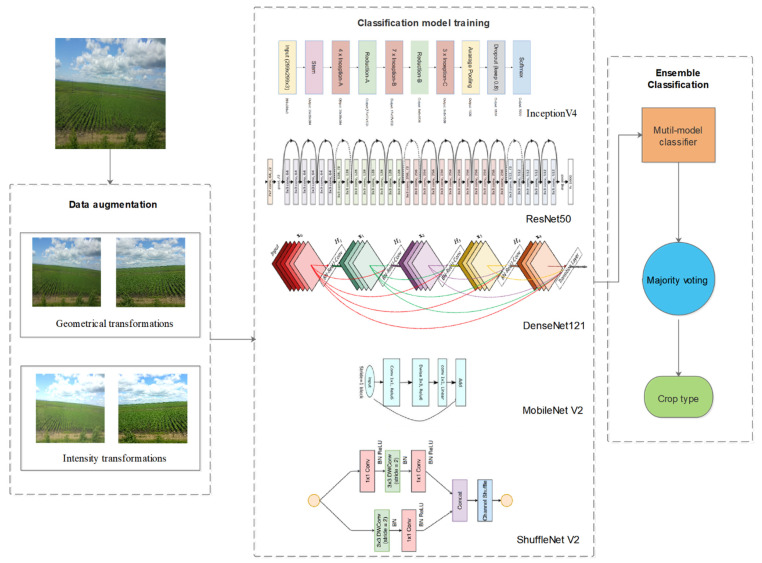
The overall model framework of CNN model for crowdsourcing crop photo classification.

**Figure 4 sensors-21-01165-f004:**
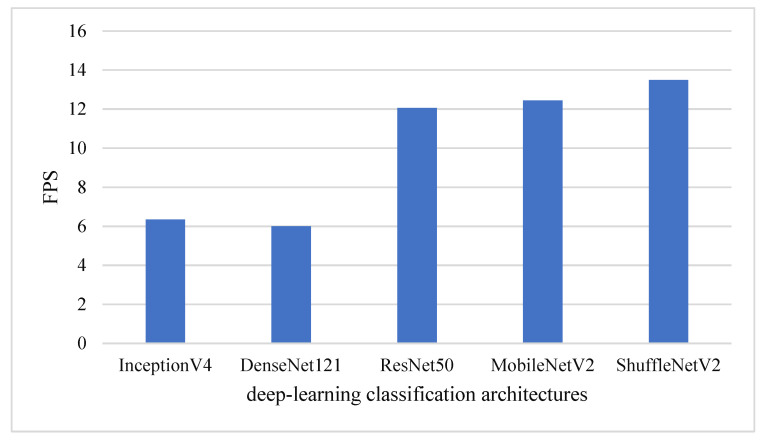
Speed performance of five deep learning classification networks.

**Figure 5 sensors-21-01165-f005:**
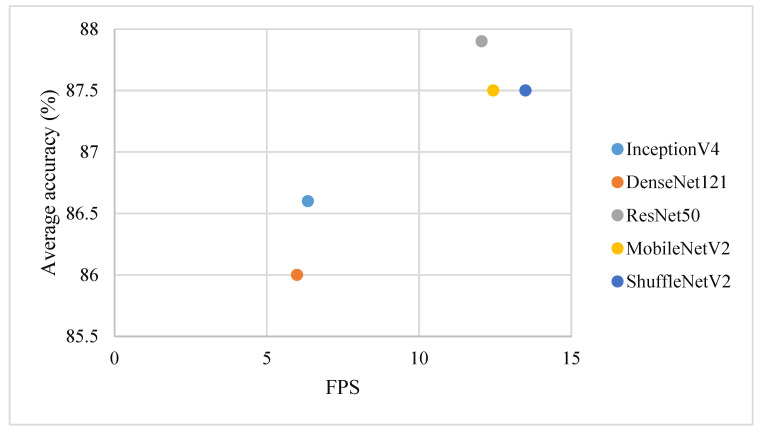
Average accuracy vs. FPS of five deep learning classification networks.

**Figure 6 sensors-21-01165-f006:**
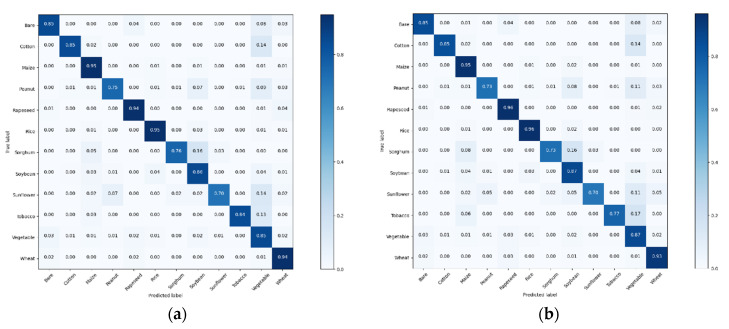
Normalized confusion matrices of the (**a**) voting-3 and (**b**) voting-5 models’ performance across the 12 cross-validated test subsets.

**Figure 7 sensors-21-01165-f007:**
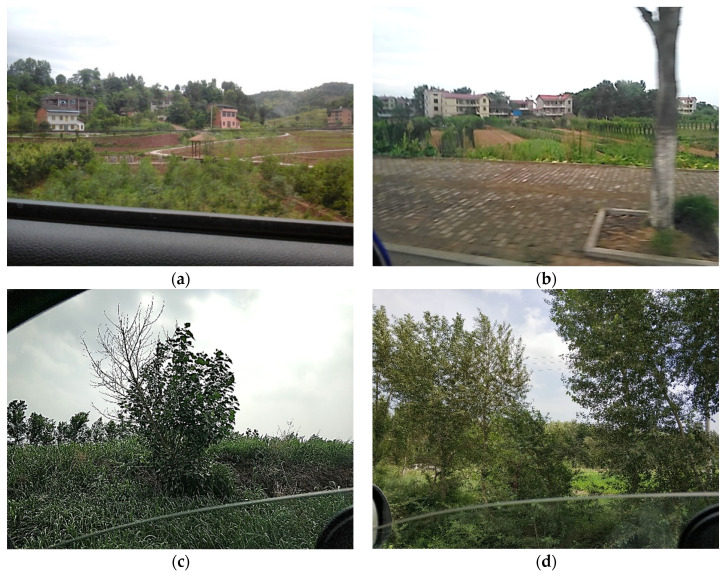
Example images highlighting confusions between classes of species. Specifically, (**a**) bare land falsely classified as vegetable, (**b**) correctly classified vegetable, (**c**) sorghum falsely classified as vegetable, (**d**) peanut falsely classified as vegetable.

**Figure 8 sensors-21-01165-f008:**
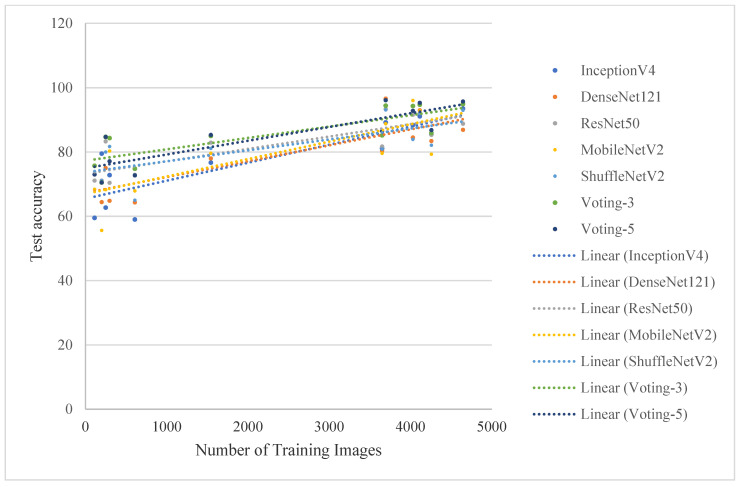
Top one-test accuracy per class against the number of training photos.

**Table 1 sensors-21-01165-t001:** Number of training and test photos in each category in the iCrop dataset.

Categories	Train	Test	Total
Bare land	1543	327	1870
Cotton	249	59	308
Maize	4114	1044	5158
Peanut	608	139	747
Rapeseed	3695	871	4566
Rice	4646	1204	5850
Sorghum	112	37	149
Soybean	4257	1080	5337
Sunflower	199	44	243
Tobacco	297	70	367
Vegetable	3651	890	4541
Wheat	4030	951	4981

**Table 2 sensors-21-01165-t002:** Test accuracy across all species computed by the five classification models.

Categories	InceptionV4	DenseNet121	ResNet50	MobileNetV2	ShuffleNetV2
Bare land	76.7	78.3	**83.2**	79.8	81.3
Cotton	62.7	76.2	**84.7**	69.5	81.4
Maize	91.1	**93.1**	92.1	92.9	92
Peanut	59.0	64.7	**73.4**	68.3	65.5
Rapeseed	89.2	**96.7**	93.5	88.9	93.2
Rice	93.5	87.0	88.9	**95.5**	93
Sorghum	59.5	73.0	73.0	70.3	**75.7**
Soybean	85.5	83.5	**85.4**	79.4	82.1
Sunflower	79.5	65.9	**72.7**	56.8	**72.7**
Tobacco	72.9	65.7	71.4	81.4	**82.9**
Vegetable	81.1	80.4	**81.8**	79.7	81.3
Wheat	88.4	84.5	91.7	**96.0**	89.3
Average accuracy	**86.2**	**86**	**87.9**	**87.5**	**87.5**

Numbers in bold represent the best classification accuracy for each cropland type.

**Table 3 sensors-21-01165-t003:** Test accuracy across all species computed by the fusion of three or five CNN classifiers.

Categories	Voting-3	Voting-5
Bare land	85	85.3
Cotton	84.7	84.7
Maize	94.7	95.3
Peanut	74.8	72.7
Rapeseed	94.4	96.1
Rice	94.9	95.7
Sorghum	75.7	73.0
Soybean	85.9	86.8
Sunflower	70.5	70.5
Tobacco	84.3	77.1
Vegetable	85.2	86.5
Wheat	94.3	92.8
Average accuracy	90.6	91.1

## Data Availability

The data presented in this study can be found here: http://www.nwatch.top:8085/icrop.

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
