# Peer review of "Identification of Crop Type in Crowdsourced Road View Photos with Deep Convolutional Neural Network"

_sensors, 2021, doi:10.3390/s21041165_

Round 1

Reviewer 1 Report

The authors present a very interesting paper in which they use public images to identify the type of crop using DCNN. They compare different methods to identify the one that provides the higher accuracy. The authors include an extended discussion, which highlight their results and demonstrate the benefits of their proposal and the knowledge of the authors in this topic. Some minor issues have to be solved before accepting the paper. Following I list these issues with the aim of enhance the quality of the paper:

  • The last paragraph of the introduction must be divided into two short paragraphs. The first one should describe the aim of the paper; while the second one describes the structure of the rest of the paper
  • A new section in which the state of the art is defined and analysed must be included. In this section, authors must show the efforts of other authors in recognizing the cultivated crops with images using AI. The authors must define the results as accuracy, number of analysed pictures etc…of other authors if possible. At the end of this section, the authors must identify the gap in the current solutions (e.g. low accuracy, small number of used pictures, too complex systems…). After the gap authors have to mention how their proposal is going to cover this gap.
  • In the references, authors should include the information of all used software or hardware. For example, detail the used smartphone and as references include its datasheet. In case of software, as GVG, include their user manual or their webpage.
  • At the end of discussion, authors should compare their results with results of similar papers. Authors must show if other authors have performed similar experiments and the accuracy of those other experiments. Then, compare if their method reached higher results than the others did or similar results with a simplified method. In summary, they must identify the progress beyond the state of the art that offers their contribution.

Author Response

Thanks for your comment. 

  • We have divided the last paragraph into two short paragraphs according to your advice in the revised version.
  • Other research recognized the cultivated crops with images from Google Street View using AI have been introduced in paragraph 5. The number of classes, overall classification accuracy, and number of analyzed pictures of other study have been given. At the end of paragraph 5, we have identified the gap in the current solutions: one is a large and public dataset is lack for application of deep learning technology and the development of intelligence in acquiring accurate crop types, another is state-of-the-art classification networks and model ensemble could be compared and selected to improve performance.
  • The GVG software can run on Android or iOS mobile platform, thus there’s a lot of hardware available. A tutorial of field data collection with GVG APP can be download from http://www.nwatch.top:8085/icrop/docs/gvg.pdf.
  • At the end of discussion, we have discussed difference of our photos with GSV. Because our photos are more challenging to classify than GSV photos and the method is also different, we can’t compare their results with results of similar papers. Instead, we compare results from different network and model ensemble with same dataset.

Reviewer 2 Report

1, Please provide API and documentation for the public to consume the established dataset. I was trying to take a look at the dataset, and did not see friendly approach to see the dataset. It took forever to log in the dataset, not sure what is the reason, can you please improve it? 

2, Regarding the network used in this paper,  please provide 

1) How are these network is implemented? pytorch? tensorflow?  trained from scratch or used some pre-trained model like ImageNet? What is the loss curve, learning rate, training epochs?  

2) Since the paper compared speed of these networks, please provide more detail of the implementations. Under what conditions are these networks are compared? Are they using the same language, like pytorch, same GPU, same size, etc. ? 

3) Please provide detailed error analysis of the results, especially for peanut, sunflower, sorghum. Why the performances for these crops are low even after fusion?  How can we improve it? Is is caused by the network design, or dataset imbalance? 

Author Response

Thanks for your comment.

1. You will find the basics about how to browse, rate, and download the established dataset in http://www.nwatch.top:8085/icrop/docs/User-guide-for-iCrop.pdf. If you just browse or rate the photos of iCrop dataset, it’s no need to log in the dataset. If you have registered and loged in, you can download all the photos.

2. Regarding the network used in this paper, we provide

  • These networks were implemented in PaddlePaddle deep learning frameworks, which is an open-sourced platform with advanced technologies and rich features . An excellent RMSprop optimizer proposed by Geoff Hinton was used in training the adaptive learning rate. Training batches of size 30 were created by uniformly sampling from all available training photos as opposed to sampling uniformly from the classes.
  • Under same conditions are these networks are compared. All models were trained and tested on an Intel(R) Xeon(R) Gold 6148 CPU @ 2.40 GHz with NVIDIA Tesla V100-SXM2 GPU and 16G RAM. During training and testing, the photo was adjusted to same size as the input of the network. Python is used as the program language to implement all the network. The test data, iteration times are the same.
  • It can be clearly seen from the figure 8 that there is a positive correlation between the number of training images and the test accuracy. The consensus of most current researches is that for deep learning, the performances will increase with growing data size. The training images of peanut, sunflower and sorghum are all lower than those of other crops, and the performances for these crops are low with or without fusion. No imbalance-correcting technique can match adding more training data when it comes to measuring precision and recall. As a result, the corresponding data collection should be carried out for the classes with fewer photo samples use a hybrid method that fuses GSV images and crowdsourcing data.

Round 2

Reviewer 2 Report

No more comments.